# A multi-environmental tracer study to determine groundwater residence times and recharge in a structurally complex multi-aquifer system

Cornelia Wilske[1,2], Axel Suckow[2], Ulf Mallast[1], Christiane Meier[3], Silke Merchel[4], Broder Merkel[5], Stefan Pavetich[4,6], Tino Rödiger[7], Georg Rugel[4], Agnes Sachse[7], Stephan M. Weise[1], Christian Siebert[1]

[1] Helmholtz Centre for Environmental Research, Dept. Catchment Hydrology, Halle/Saale, 06120, Germany.
[2] CSIRO Land and Water, Urrbrae (SA), 5064, Australia
[3] UBA Umweltbundesamt, Dessau-Roßlau, 06844, Germany
[4] Helmholtz-Zentrum Dresden-Rossendorf, Dresden, 01328, Germany
[5] TU Bergakademie Freiberg/Sachs., 09599, Germany
[6] Australian National University, Research School of Physics and Engineering, Dept. Nuclear Physics, Canberra, ACT 2609, Australia
[7] Helmholtz Centre for Environmental Research, Dept. Computational Hydrology, Leipzig, 04318, Germany

*Correspondence to*: Christian Siebert (Christian.siebert@ufz.de)

**Abstract.** Despite being the main drinking water resource for over five million people, the water balance of the Eastern Mountain Aquifer system on the western side of the Dead Sea is poorly understood. The regional aquifer consists of fractured and karstified limestone — aquifers of Cretaceous age and can be separated in Cenomanian aquifer (upper aquifer) and Albian aquifer (lower aquifer). Both aquifers are exposed along the mountain ridge around Jerusalem, which is the main recharge area. From here, the recharged groundwater flows in a highly karstified aquifer system towards the east, to discharge in springs in the Lower Jordan Valley and Dead Sea region. We investigated the Eastern Mountain Aquifer system on groundwater flow, groundwater age and potential mixtures, and groundwater recharge. We combined $^{36}$Cl/Cl, tritium and the anthropogenic gases SF$_6$, CFC-12 and CFC-11, CFC-113 as "dating" tracers to estimate the young water components inside the Eastern Mountain Aquifer system. By application of lumped parameter models, we verified young groundwater components from the last 10 to 30 years and an admixture of a groundwater component older than about 70 years. Concentrations of nitrate, Simazine® (Pesticide), Acesulfame K® (artificial sweetener) and Naproxen® (drug) in the groundwater were further indications of infiltration during the last 30 years. The combination of multiple environmental tracers and lumped parameter modelling helped to understand the groundwater age distribution and to estimate recharge despite scarce data in this very complex hydrogeological setting. Our groundwater recharge rates support groundwater management of this politically difficult area and can be used to inform and calibrate ongoing groundwater flow models.

## 1 Introduction

About 20 percent of the Earth's land surface is covered by carbonate-karst or sulphate aquifers and serve as the primary water resource for at least 25% of the world's population. In addition, about a fifth of the world's karst systems are in (semi-) arid areas, whose water scarcity is aggravated by strong population growth (Ford and Williams, 2007). Karst systems represent abundant, but highly variable water resources whose extremely heterogeneous and anisotropic flow behaviour prevents exact predictions regarding mass transport and the usable water quantities (Bakalowicz, 2005). Nevertheless, the water balance stays the basic requirement for sustainable management and protection of any water resources.

Environmental tracers play an important role in sustainable water management strategies because they allow to estimate the groundwater age distribution with depth and together with simple lumped parameter models to quantify groundwater infiltration rates (Vogel, 1967; Solomon et al., 1995; IAEA, 2006). Especially in karst aquifers, wide ranges of residence times are observable due to the strongly heterogeneous hydraulic system, allowing water to rapidly flow through conduits and fractures and very slowly flow through the small pores of the matrix. That leads to large heterogeneities in the groundwater age distribution requiring hence especially in karst the application of multiple tracers to constrain the age distribution. We define young groundwater as having measurable concentrations of anthropogenic tracers and therefore a mixing component recharged after about 1950, while in old groundwater these tracers are not detectable (e.g. Plummer et al. 1993, Cook & Herczeg 2000, Hinsby et al. 2001a). The atmospheric tracer CFC-11, CFC-12, CFC-113, SF6 (sulphur hexafluoride) and $^{36}$Cl/Cl and tritium from bomb-tests or anthropogenic organic trace pollutants like pesticides, sweeteners or drugs are increasingly used as tracers of young groundwater (IAEA 2006). Gas tracers like CFCs and SF6 move through the unsaturated zone primary by diffusion, leading to a time lag at the water table compared to the atmosphere (Cook and Solomon, 1995; Cook et al., 1995). A time lag is also possible for the water bound tracers tritium and $^{36}$Cl, since the advection through the unsaturated zone may take decades (Suckow et al., 1993; Lin and Wei, 2006) and in infiltration areas dominated by sand or clay, water bound tracers are generally slower than gas tracers (Solomon et al., 1992; Cook et al., 1995). This can be very different in karst systems, where preferential flow in karst "tubes" allows fast recharge to the groundwater table and fluctuations of groundwater level may allow further gas exchange thereafter.

The deconvolution of measured tracer concentrations into recharge rates therefore needs modelling. If age of water would be known as function of depth, any flow model could be directly constrained and the recharge rates deduced. However, the "idealized groundwater age", which is often understood as the time span an imaginary water parcel needs between infiltrating at the groundwater surface and being sampled at a well or spring (Suckow, 2014a) is not directly measurable. In addition, groundwater mixes both along its natural flow through the aquifer and during sampling in the well. Therefore, simple lumped parameter models (LPM) are used to interpret the measured tracer concentrations as mean residence times (MRT) via a convolution integral, which in combination with the underlying assumptions on the flow system allow deducing recharge rates.

We applied the piston flow model (PM), the dispersion model (DM) and the partial exponential model (PEM) to approximate the age distribution in our groundwater samples.

In our study, the Eastern Mountain Aquifer system (EMA) in the western Dead Sea catchment is the pivotal water resource for some million people in the Westbank and Israel. Unequal distribution of both borehole information lead to poor and limited data for studying that aquifer system. Previous studies considered age dating tracers to quantify water movement and flow velocity within the EMA and associated aquifers. Paul et al. (1986) and Yechieli et al. (1996) studied $^{36}Cl/Cl$ to detect very old groundwater brines in the Dead Sea area. In contrast young-age dating tracers such as tritium or anthropogenic trace gases (CFCs and SF6) were used to quantify the duration of water flow from recharge areas to the springs (Lange 2011). Environmental tracer investigations of the main Cretaceous aquifers (Upper Cenomanian and Albian) in the western Dead Sea catchment attempted to quantify the duration of water flow from the recharge area to the springs in the mountain region uphill the Dead Sea coast. The young-age dating tracers demonstrated a large young groundwater fraction with a mean residence time of less than 30 years in the springs of the mountain region and fast connections to the recharge area. All previous studies together show large heterogeneities in the groundwater age distribution (Avrahamov et al., 2018).

In this study we combine for the first time in this area bomb-derived $^{36}Cl$, anthropogenic organic trace substances and environmental tracers like tritium, CFCs, SF6 in combination with lumped parameter models to interpret the distribution of these tracers to quantify recharge. Maloszewski & Zuber (1982, 1993, 1996) have shown that LPM are a useful tool for interpreting tracer data obtained at separate sampling sites when it is not possible to use distributed parameter models, as the latter require more detailed and often unavailable knowledge of distributed parameters for the investigated system. In detail this work aims to (i) validate young rainwater input and short groundwater travel times via karst conduits, related to rapid flow paths from the recharge area; (ii) to quantify the time lag of gas tracers in the unsaturated zone; (iii) to quantify groundwater mixing of groundwater components with different age via lumped parameter models and (vi) to estimate groundwater recharge and support calculations of future groundwater resource development.

## 1.1 Study Area

The study area, which represents the western surface drainage basin of the Dead Sea, is embedded in a region that is morphologically and geologically dominated by the tectonic processes associated with the Jordan-Dead Sea Rift, being active since the late Oligocene (Garfunkel et al., 1981; Rosenfeld and Hirsch, 2005). The western rift fault separates the Cretaceous aquifer formations that form the Graben shoulder from the deeply subsided Graben and its Quaternary filling. In addition, rift tectonics induced a series of faults within the western Graben Shoulder, resulting in down-faulted blocks, which find their surface expression in a strong morphological gradient. Within less than 25 km, the land surface drops from +800 m msl. in the west to sea level at the rift margin and with a terminal step to -430 m msl. at the Dead Sea in the east (Fig. 1).

The semi-arid to arid Mediterranean climate leads to precipitation during the winter season, but with a strong decline from west to east due to which the study area can be divided into three hydrological zones: (i) the recharge area in the upland that receives annually up to 580 mm of precipitation; (ii) the transition zone occupying the hillsides of the upland down to the rift

95    margin, receiving 100-400 mm/a and (iii) the major discharge area of groundwater in the Lower Jordan Valley/Dead Sea area, receiving less than 100 mm/a of precipitation.

The Graben shoulder hosts a thick aquifer system that is mainly built of fractured and in layers karstified Upper Cetaceous lime- and dolostones, which are overlain by a Senonian chalky aquitard (Kronfeld and Rosenthal 1987; Weinberger and Rosenthal 1996). A marly aquiclude (Lower Cenomanian) divides the system into a Lower Aquifer (Albian) and Upper Aquifer

100    (Upper Cenomanian). Occurring impervious beds in the Upper Aquifer permit the development of a perched and locally important aquifer (Turonian), being built of homogenous and fissured limestones and holds springs, which emerge in deeply incised valleys within the transition zone, (e.g. Wadi Qilt springs; Fig. 2). While all aquifer units are recharged in the mountain area, the natural discharge of the two regional aquifers occurs through springs at the base of the Graben Shoulders, where the groundwater leave the aquifers and approach the prevalently impervious Quaternary Graben filling.

105    Subsequently, groundwater emerges along the shore in spring clusters, forming ecologically important oasis such as Ein Feshkha. Where groundwater can percolate into the Quaternary sediments, they intensely dissolve the contained evaporite minerals (halite, anhydrite, aragonite) and get saline on their flow path to the lake shore. In addition, the chemical and isotopic composition of approaching fresh groundwaters get systematically modified by admix of brines, already in the vicinity of the major rift fault (Katz and Kolodny, 1989; Stein et al., 1997; Ghanem, 1999; Yechieli, 2000; Klein-BenDavid et al., 2004;

110    Khayat et al., 2006a; Khayat et al., 2006b; Möller et al., 2007; Siebert et al., 2014; Starinsky and Katz, 2014). Groundwater recharge rates are controlled by climate conditions and have been investigated earlier to force regional groundwater models (Guttman et al., 2004; Yellin-Dror et al., 2008; Gräbe et al., 2013; Schmidt et al., 2014), which allowed a detailed insight of the regional groundwater flow dynamics (Laronne Ben-Itzhak and Gvirtzman 2005; Sachse 2017).

Human groundwater abstraction takes place mainly in the mountain ridge inside the recharge area and along the transition to

115    the Jordan Valley. This unequal distribution of the sampling possibilities led to a data scarcity for the entire central area of the aquifer, which is also visible in Figure 1.

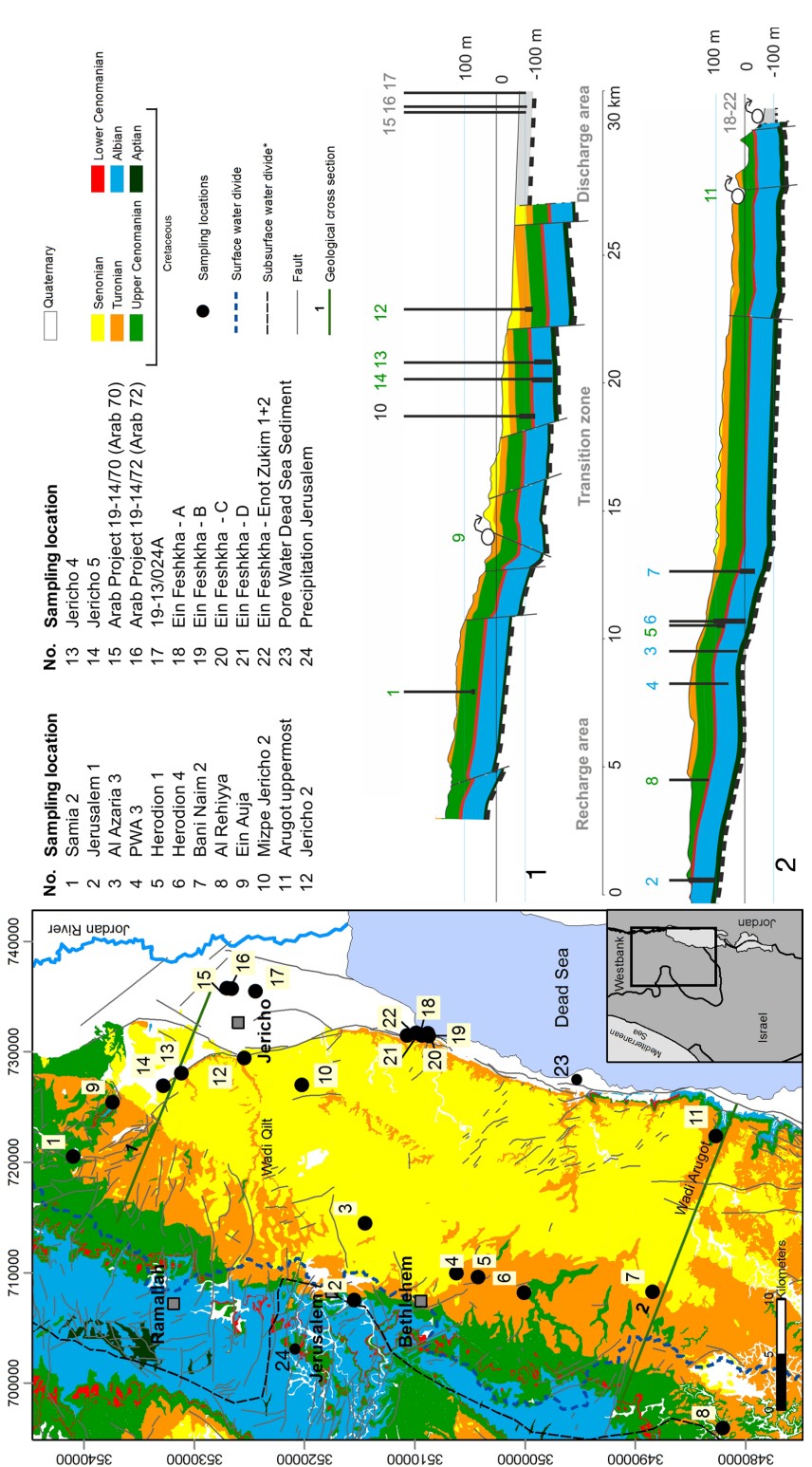

**Figure 1: shows the location of the study site including geological information based on (Begin, 1974; Mor and Burg, 2000; Raz, 1986; Roth, 1973; Shachnai, 2000; Sneh and Avni, 2011; Sneh and Roth, 2012). Coordinates: UTM WGS84 Z36N.**

## 2 Materials and Methods

### 2.1 Fundamentals of the Method

Age distributions of young groundwater can be characterized by applying anthropogenic trace gases like CFC-11, CFC-12, CFC-113 and SF6 in lumped parameter models if an input function is available. For the last 6 decades, that function can be derived for gas tracers from (i) their known concentrations in the atmosphere (Fig. 2), (ii) the observation that they are well mixed in the atmosphere, and (iii) that their solubility at the temperature of recharge is known from Henry's Law (Plummer & Busenberg 1999). The large-scale production of CFC-11 and CFC-12 (both used as cooling fluid) started in the early 1940s while production of CFC-113 started in 1960s only. Inevitably they leaked into the environment, with atmospheric concentrations rising until the 1990s, when their moratorium took effect. Another industrial gas SF6, widely used as electrical insulator is detectable in the atmosphere since 1960s with still exponentially rising concentrations. Atmospheric concentrations of $SF_6$ and the CFCs for the period 1953-2006 (Fig. 2) are derived from Plummer et al. (2006). Since gas solubility in recharging precipitation depends on temperature and atmospheric pressure, the average air temperature of the winter season in the study area (15°C) was used for the former and an altitude of 700 m msl., which is the average altitude of the infiltration area, was used for the latter. The rain water salinity was set to 0 [‰] since rain water is low mineralised.

Tritium ($^3H$) is the naturally-occurring isotope of hydrogen and is mainly produced by fast secondary neutrons from cosmic radiation. It decays to $^3He$ with a half-life time of $12.32 \pm 0.02$ a (Lucas and Unterweger, 2000), making $^3H$ usable for groundwater age dating in a time frame of <40 years (e.g. (Schlosser et al., 1988; Solomon et al., 1992; Cook and Solomon, 1997; Sültenfuß and Massmann, 2004), including quantifying changes of measured $^3He/^4He$ ratios in groundwater. At the study area, observation of tritium in precipitation (GNIP stations Beer Sheva, Bet Dagan and Tirat Yael (IAEA/WMO, 2019)) were available for 1960-2001 only, while input data are required until 2014. We therefore applied tritium data of Vienna station, Austria (IAEA/WMO, 2019), which are adjusted to the longitudinal and latitudinal difference by a factor of 0.4 to match the Israel stations (Fig. 2). For the synoptic tracer plots the decay correction is to October 31, 2013. The pre-bomb input value for tritium was set to a mean tritium concentration of 3 TU obtained from the GNIP data base (IAEA/WMO, 2019).

$^{36}Cl$ is produced naturally via cosmic-ray and solar protons induced nuclear reactions of argon in the atmosphere and of $^{35}Cl$ in marine aerosols (Alvarado et al., 2005). However, comparable to $^3H$, the atmospheric concentration of "bomb" $^{36}Cl$ peaked during the 1950s as an effect of nuclear weapon tests and was washed out from the atmosphere until the end of the 1960s. The $^{36}Cl$ bomb peak precedes the tritium peak by half a decade. The $^{36}Cl$ input curve for our study area (Fig. 2) was obtained from Iceland ice core measurements from Synal et al. (1990), which were corrected to the location of the study area applying a latitudinal correction with a factor of three, according to (Heikkilä et al. (2009), who modelled $^{36}Cl$ fall-out for different latitudes. Natural "pre-bomb" concentration of $^{36}Cl/Cl$ was assumed to be $10^{-14}$, which is based on $^{36}Cl/Cl$ in rainwater, sampled during winter 2014/2015 with an average value of $^{36}Cl/Cl=8\times10^{-15}$.

Most studies to estimate groundwater age with $^{36}$Cl base on the half-life of $^{36}$Cl (0.301±0.015 Ma) (Nica et al., 2006) and consider time frames >100,000 years (Davis et al., 1983; Bentley et al., 1986; Love et al., 2000; Mahara et al., 2012; Müller et al., 2016). Studies using $^{36}$Cl from the atmospheric bomb peak to estimate groundwater age of the last decades are much less-frequent (Alvarado et al., 2005; Tosaki et al., 2007; 2010; Lavastre et al., 2010; Rebeix et al., 2014).

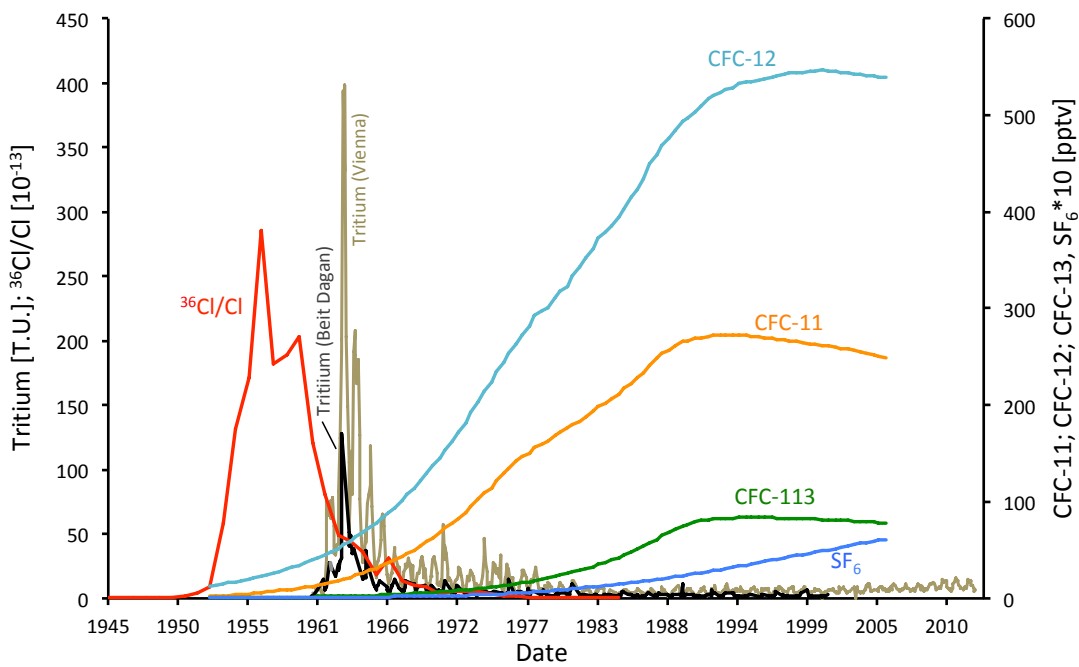

**Figure 2: Atmospheric input curves of $^{36}$Cl/Cl ($^{36}$Cl obtained from Dye-3 ice core, Synal et al. (1990)), SF$_6$ and CFCs (Plummer et al., 2006) and decay-corrected Tritium in rain water of Beit Dagan and Vienna (IAEA/WMO, 2019).**

Anthropogenic organic trace pollutants in groundwater are associated to nutrition, medication or agricultural and industrial

development. During the last decades, artificial sweeteners played an important role as a surrogate in the nutrition industry. Particularly Acesulfame K® (ACE-K) is used since the 1990s and is stable against waste water treatment (WWT) processes, making it an ideal tracer for domestic wastewater. A second chemical marker for human intake is Naproxen® (NAP), widely applied as anti-inflammatory drug since the 1980s. Though NAP is partly eliminated during WWT and may be adsorbed along flow-paths to sediments (Chefetz et al., 2008; Yu, et al., 2008; Teijón et al., 2013), it occurs in effluents of sewage plants and

increasingly in natural waters (Arany et al., 2013). Contrasting that, ACE-K is hydrophilic (e.g. Buerge & Poiger, 2011) and inert against degradation, which makes it a valuable substance to trace transport from the recharge- towards the discharge area. In addition to the urban indicators, pesticide traces in groundwater were used to identify the agricultural contributions to the water resources. Simazine®, which is available since the 1950s is one of the most applied herbicides and absorbs to soil where it may be eliminated though bacterial degradation. However, the use of Simazine® is phased out in Israel since 2012-2014

(Berman et al., 2014). Nitrate is another indicator for anthropogenic input, whose distribution in groundwater originates from nitrification of $NH_4$ during WWT and as a fertilizer excess in agriculture.

## 2.1 Sampling and analytical methods

In the study area, 22 groundwater samples were taken from springs and active wells (sampling locations in Fig. 1, Tab. 1) after reaching stable conditions for temperature, electrical conductivity and pH. Major anion samples were filled into HD-PE bottles
after passing a 0.45 µm cellulose acetate filter. Samples for cations were acidified using $HNO_3$. Samples for $^{36}Cl$ analyses are filled into 500 ml HD-PE bottles, which have been specifically pre-cleaned using ultrapure $HNO_3$. Samples for $^{36}Cl$ analyses were acidified with $HNO_3$. Following the methodology described in Oster et al. (1996), sampling for CFCs and $SF_6$ was performed using glass bottles fully submerged in tins, filled with sampling water. Tritium samples were collected in 500 ml HD-PE bottles. Organic trace elements were sampled in pre-cleaned, methanol flushed 1000 ml brown glass bottles with the
use of a glass microfiber filter (0.7 µm).

Major element analyses in water samples were performed at the Helmholtz-Centre for Environmental Research (UFZ) by using matrix adjusted ICP-AES (Spectro Arcos) for $Na^+$, $K^+$, $Ca^{2+}$, $Mg^{2+}$ and $Sr^{2+}$ and by using ion chromatography (Dionex ICS-2000) for $Cl^-$, $Br^-$ and $SO_4^{2-}$. Bicarbonate was determined *in situ* by titration. Analyses of $^{36}Cl$ were carried out in Helmholtz-Zentrum Dresden-Rossendorf at the accelerator mass spectrometry (AMS) facility DREsden AMS (DREAMS) (Akhmadaliev
et al., 2013). The main preparation steps (Conard et al., 1986) consist of: (i) precipitation of chloride via adding $AgNO_3$ solution (10%) and subsequent dissolution of the AgCl in $NH_{4aq}$; (ii) separation of chlorides from sulphates by co-precipitation of $BaSO_4$ with $BaCO_3$ ($CO_2$ from air) using saturated $BaNO_3$ solution and filtration through a syringe filter made of polyvinylidene fluoride (pore size: 0.45 µm); (iii) re-precipitation of AgCl in the filtrate with nitric acid. $^{36}Cl$ is measured with AMS relative to the stable Cl isotopes, $^{35}Cl$ and $^{37}Cl$ (Pavetich et al., 2014; Rugel et al., 2016), and is given as ratio
$^{36}Cl/(^{37}Cl+^{35}Cl)$ (termed $^{36}Cl/Cl$ in the text). All data is normalized to the standards SM-Cl-12 and SM-Cl-13 (Merchel et al., 2011).

Tritium sample preparation and measurement were conducted by the isotope hydrology group at UFZ, following the preparation steps of Trettin et al. (2002) by first enriching a 400 ml water sample electrolytically and measure it via liquid scintillation counting with a detection limit of 0.5 TU.
CFCs and $SF_6$ were analysed in the Spurenstofflabor Dr. Harald Oster (Wachenheim, Germany) by Gas Chromatography (Bullister and Weiss, 1988; Oster et al., 1996). Measurements of organic trace elements were conducted with High Pressure Liquid Chromatography-Mass Spectrometry (HPLC-MS) at the UFZ. To transport the samples to the lab, organic components were stabilized by solid phase extraction (SPE) on cartridges containing a polar-modified polystyrene-divinylbenzene copolymer (Chromabond Easy, Machery-Nagel). At the lab, analytes were eluted with methanol and measured with HPLC-
MS/MS (Agilent 1000, Agilent Technologies, Germany; coupled with an API2000 mass spectrometer (AB Sciex, Germany)). Limits of detections are 1 µg/l, 0.6 µg/l and 0.3 µg/l for Naproxen®, Acesulfame K® and Simazine®, respectively. Analytical Results are given in Table 1.

## 2.1 Lumped parameter model "LUMPY"

LPM are pre-defined analytical solutions of simplified flow systems. They describe the tracer output mathematically with a convolution integral that combines the tracer input history, weighed with the age distribution valid for the flow system in question (Maloszewski and Zuber, 1982; 2002).

Mathematically all lumped parameter models for steady-state flow systems with a time-variable tracer input are convolution integrals (Equation 1):

$$C_{out}(t) = \int_0^\infty C_{in}(t - t') \exp(-\lambda t') \, g(t') dt' \tag{1}$$

where t = calendar time; t' = transit time of the tracer; $C_{out}$ = output concentration; $C_{in}$ = input concentration; and g(t') = weighting function or system response function. All weighting functions of all models are normalized, following Equation (2):

$$\int_0^\infty g(t') dt' = 1 \tag{2}$$

The mean residence time (MRT) is the main fitting parameter, while some models (e.g. dispersion model) require additional parameters like (i) Péclet number, (ii) top and bottom of screened section or (iii) saturated thickness of the aquifer. Input data are (i) regional atmospheric tracer input curves and (ii) selected hydrogeological characteristics such as the infiltration temperature and elevation.

In this study the LPM code LUMPY (Suckow, 2012) was used to implement the convolution integral for flow systems that can be described by the piston flow model (PM), the exponential model (including the partial exponential model PEM) and the dispersion model (DM).

The PM describes the movement of a water parcel along a defined flow path from the aquifer surface towards the spring or well filter, neglecting any mixing, dispersion or diffusion. The DM characterizes transport influenced by dispersion and advective flow. The relative magnitude of both is expressed as the Péclet number Pe (Equation 3):

$$Pe = l \cdot v / D \tag{3}$$

where $l$ is the flow length of the system under consideration, $v$ is the velocity and $D$ is the dispersion constant (Huysmans and Dassargues, 2005). In our model approach, best fits were obtained by applying a Péclet number of 30, characterizing dominantly advective transport.

# Table 1. sampling locations and analytical results.

| No | Station | ID | Sampling date | x utm | y utm | Elevation [m NN] | USZ [m] | pH | Eh [mV] | T[°C] | EC [µS/cm] | $NO_3$ [mg/l] | Cl [mg/l] | $^{36}Cl/Cl$ $[10^{-15}]$ | CFC-11 [pmol/l] | CFC-12 [pmol/l] | CFC-113 [pmol/l] | $SF_6$ [fmol/l] | Tritium [T.U.] | Simazine [ng/l] | Acesulfame K [ng/l] | Naproxene [ng/l] |
|---|---|---|---|---|---|---|---|---|---|---|---|---|---|---|---|---|---|---|---|---|---|---|
| *Wells - recharge zone (Cenomanian)* | | | | | | | | | | | | | | | | | | | | | | |
| 1 | Samia 2 | 12513 | 10.11.2012 | 720608 | 3541490 | 420.3 | 17.5 | 7.5 | 429 | 22.3 | 476 | 16.5 | 38.0 | 1.57 ± 0.08 | 2.3 ± 0.3 | 1.2 ± 0.1 | 0.19 ± 0.05 | 1.0 ± 0.2 | 2.1 ± 0.5 | 3.8 | 0.4 | |
| | Samia 2 | 13504 | 22.05.2013 | | | | | 7.5 | 349 | 23.6 | 479 | 16.3 | 34.8 | | | | | | | 4.9 | 13.7 | 137.4 |
| 5 | Herodion 1 | 12505 | 08.11.2012 | 710448 | 3504543 | 596.7 | 254.15 | 7.5 | 431 | 21.0 | 491 | 14.7 | 27.6 | 1.41 ± 0.07 | 1.7 ± 0.2 | 0.73 ± 0.05 | 0.1 ± 0.05 | 1.0 ± 0.2 | 1.2 ± 0.3 | | 0.2 | |
| 8 | Al Rehiyya | 13515 | 24.05.2013 | 697253 | 3482073 | 698.2 | | 7.3 | 352 | 25.5 | 599 | 4.71 | 44.1 | 0.17 ± 0.02 | | | | | | 4.4 | | 5.6 |
| *Wells - recharge zone (Albian)* | | | | | | | | | | | | | | | | | | | | | | |
| 2 | Jerusalem 1 | 13497 | 10.02.2013 | 708126 | 3515719 | 701.0 | 120.9 | 7.3 | 280 | 20.4 | 679 | 19.5 | 64.3 | 8.91 ± 0.40 | | | | | 1.7 ± 0.3 | 148.5 | 5.6 | 168.8 |
| | Jerusalem 1 | 13505 | 23.05.2013 | 708126 | 3515719 | | | 7.2 | 305 | 20.6 | 692 | 25.2 | 67.4 | | | | | | | 4.4 | 13.0 | |
| *Wells - transition zone (Albian)* | | | | | | | | | | | | | | | | | | | | | | |
| 3 | Al Azaria 3 | 12507 | 09.11.2012 | 715119 | 3514901 | 497.6 | | 7.3 | 424 | 24.1 | 607 | 21.6 | 45.0 | 0.96 ± 0.06 | ca. 180 | 5.2 ± 0.3 | 0.9 ± 0.1 | 2.9 ± 0.3 | <0.5 | 2.2 | | |
| | Al Azaria 3 | 12511 | 24.05.2013 | 715119 | 3514901 | | | 7.3 | 335 | 25.0 | 569 | 16 | 35.8 | | | | | | | 1.7 | 7.5 | 62.3 |
| 4 | PWA 3 | 13513 | 24.05.2013 | 710746 | 3506529 | 629.5 | | 7.4 | 346 | 24.6 | 514 | 7.95 | 22.7 | 0.49 ± 0.03 | | | | | | 1.5 | | 6.5 |
| 6 | Herodion 4 | 13510 | 23.05.2013 | 709107 | 3500349 | 774.0 | 458.9 | 7.2 | 350 | 24.1 | 548 | 4.86 | 23.2 | 0.32 ± 0.03 | | | | | | 5.5 | | 66.1 |
| 7 | Bani Naim 2 | 12510 | 09.11.2012 | 709445 | 3488704 | 540.5 | 463.5 | 7.3 | 415 | 26.7 | 551 | 5.64 | 22.7 | 0.19 ± 0.02 | 0.02 ± 0.05 | <0.01 | <0.01 | 0.5 ± 0.1 | <0.5 | 0.4 | | |
| *Wells - transition zone (Cenomanian)* | | | | | | | | | | | | | | | | | | | | | | |
| 10 | Mitzpe Jericho 2 | 12518 | 12.11.2012 | 727516 | 3520896 | -19.8 | 328.71 | 7.3 | 509 | 24.5 | 758 | 43.3 | 74.4 | 0.80 ± 0.05 | 25 ± 5 | 2.7 ± 0.2 | 4 ± 1 | 0.1 ± 0.1 | 1.4 ± 0.3 | 1.4 | 3.3 | |
| | Mitzpe Jericho 2 | 12520 | 28.05.2013 | 727516 | 3520896 | | | 7.2 | 377 | 24.7 | 761 | 42.7 | 73.5 | | | | | | | 9.2 | 22.1 | 0.0 |
| *Springs - transition zone N (Cenomanian)* | | | | | | | | | | | | | | | | | | | | | | |
| 9 | Ein Auja | 13470 | 04.02.2013 | 725617 | 3538000 | 32.0 | | 7.2 | 387 | 20.9 | 618 | 21 | 39.0 | 0.64 ± 0.04 | | | | | 2.9 ± 0.3 | 25.7 | 6.4 | |
| | Ein Auja | 13518 | 27.05.2013 | 725617 | 3538000 | | | 7.2 | 500 | 21.3 | 561 | 15.9 | 34.5 | | | | | | | | 33.3 | 11.8 |
| *Springs - transition zone S (Cenomanian)* | | | | | | | | | | | | | | | | | | | | | | |
| 11 | Arugot uppermost | 12341 | 27.10.2012 | 723664 | 3483268 | -152.9 | 328.71 | 7.3 | 413 | 27.3 | 835 | 19 | 117.0 | | 2.5 ± 0.3 | 1.4 ± 0.1 | 0.21 ± 0.05 | 1.8 ± 0.2 | 0.5 ± 0.3 | | 0.4 | 1.3 |
| *Wells - discharge zone (Cenomanian)* | | | | | | | | | | | | | | | | | | | | | | |
| 12 | Jericho 2 | 13479 | 07.02.2013 | 729844 | 3526155 | -168.6 | | 7.1 | 290 | 24.7 | 1307 | 12.9 | 234 | 0.20 ± 0.02 | | | | | 2.0 ± 0.3 | 73.0 | 4.4 | |
| | Jericho 2 | 13522 | 28.05.2013 | 729844 | 3526155 | | | 7.0 | 295 | 25.1 | 1805 | <46 | 361 | | | | | | | 129.3 | 135.8 | 11.8 |
| *Wells - discharge zone (Albian)* | | | | | | | | | | | | | | | | | | | | | | |
| 13 | Jericho 4 | 13523 | 28.05.2013 | 728379 | 3531816 | -115.5 | 368.39 | 7.2 | 335 | 27.3 | 1067 | 6.6 | 208 | 0.15 ± 0.01 | | | | | <0.6 (0.3/07) | 7.2 | 56.4 | 13.5 |
| 14 | Jericho 5 | 13524 | 28.05.2013 | 727200 | 3533467 | -44.8 | 456.6 | 7.1 | 339 | 25.1 | 1369 | <46 | 224 | 0.10 ± 0.02 | | | | | <0.5 (0.3/07) | 6.2 | 14.3 | 13.7 |
| *Wells - discharge zone (Quaternary)* | | | | | | | | | | | | | | | | | | | | | | |
| 15 | Arab Project 19-14/70 (Arab 70) | 13497 | 16.02.2013 | 736108 | 3527441 | -308.7 | | 6.9 | 308 | 26.3 | 4860 | 22.2 | 1303 | 0.24 ± 0.03 | | | | | 0.4 ± 0.3 | 0.3 | 2.0 | |
| | Arab Project 19-14/70 (Arab 70) | 13526 | 01.06.2013 | 736108 | 3527441 | | | 6.9 | 357 | 26.2 | 4300 | <46 | 1202 | | | | | | | | 12.9 | 26.3 |
| 16 | Arab Project 19-14/72 (Arab 72) | 13498 | 16.02.2013 | 736124 | 3527875 | -306.8 | | 7.3 | 246 | 26.6 | 3300 | 22.6 | 926 | 0.26 ± 0.04 | | | | | | 3.4 | 1.2 | |
| | Arab Project 19-14/72 (Arab 72) | 13527 | 01.06.2013 | 736124 | 3527875 | | | 7.1 | 291 | 27.0 | 4040 | <46 | 1086 | | | | | | | 2.8 | 7.8 | 34.5 |
| 17 | 19-13/024A | 13500 | 16.02.2013 | 735921 | 3525245 | | | 7.3 | 463 | 24.8 | 1527 | 10.7 | 234 | 0.23 ± 0.02 | | | | | <0.6 | 0.7 | 0.5 | |
| *Springs - discharge zone (Quaternary)* | | | | | | | | | | | | | | | | | | | | | | |
| 18 | Ein Feshkha - A | 12354 | 01.11.2012 | 732218 | 3510111 | | | 7.3 | 337 | 26.3 | 12620 | 1.54 | 4428 | 0.11 ± 0.02 | | | | | 0.9 ± 0.3 | 0.3 | | |
| 19 | Ein Feshkha - B | 12355 | 01.11.2012 | 732412 | 3509538 | | | 7.8 | 364 | 26.8 | 5860 | 8.18 | 1935 | 0.10 ± 0.01 | | | | | 0.7 ± 0.3 | | | |
| 20 | Ein Feshkha - C | 13489 | 11.02.2013 | 732195 | 3509480 | | | 6.9 | 520 | 27.6 | 4680 | 4.13 | 1536 | 0.13 ± 0.02 | | | | | 0.8 ± 0.3 | 3.8 | | |
| | Ein Feshkha - C | 13531 | 07.06.2013 | 732195 | 3509480 | | | 7.0 | 534 | 27.5 | 4670 | <46 | 1381 | | | | | | | 0.3 | 11.3 | 40.0 |
| 21 | Ein Feshkha - D | 13476 | 05.02.2013 | 732432 | 3510612 | | | 7.5 | 107 | 24.6 | 4230 | <2.3 | 1326 | 0.15 ± 0.03 | | | | | 0.8 ± 0.3 | 3.3 | 2.9 | |
| | Ein Feshkha - D | 13502 | 22.05.2013 | 732432 | 3510612 | | | 7.8 | 244 | 26.5 | 4250 | <46 | 1194 | 0.15 ± 0.03 | | | | | | | 12.5 | 170.3 |
| 22 | Ein Feshkha - Enot Zukim 1+2 | 12356 | 02.11.2012 | 732201 | 3511443 | | | 6.8 | 418 | 29.2 | 3650 | 20.45 | 1112 | 0.15 ± 0.02 | | | | | 1.0 ± 0.3 | 1.0 | | |
| | Ein Feshkha - Enot Zukim 1+2 | 13474 | 05.02.2013 | 732201 | 3511443 | | | 7.2 | 444 | 25.2 | 4040 | 9.02 | 1058 | | | | | | 0.7 ± 0.3 | 0.4 | | |
| | Ein Feshkha - Enot Zukim 1+2 | 13532 | 07.06.2013 | 732201 | 3511443 | | | 7.2 | 392 | 28.1 | 3720 | <46 | 965 | 0.10 ± 0.02 | | | | | | 8.9 | 29.9 | 35.7 |
| *Pore water from Dead Sea sediments* | | | | | | | | | | | | | | | | | | | | | | |
| 23 | Pore water A | 13600 | 21.11.2013 | 733170 | 3510300 | | | 5.4 | | 35.6 | 182700 | | 14754 | 1.16 | | | | | | | | |
| | Pore water B | 13601 | 21.11.2013 | 733170 | 3510300 | | | 5.4 | | 35.6 | 182700 | | 237441 | 1.08 | | | | | | | | |
| | Pore water C | 13602 | 21.11.2013 | 733170 | 3510300 | | | 5.4 | | 35.6 | 182700 | | 238248 | 1.41 | | | | | | | | |
| *Precipitation* | | | | | | | | | | | | | | | | | | | | | | |
| 24 | precipitation Jerusalem | 14517 | 24.11.2014 | 709690 | 3519284 | | | 6.2 | | | | | 6.4 | 0.13 ± 0.02 | | | | | | | | |
| | precipitation Jerusalem | 14518 | 26.11.2014 | 709690 | 3519284 | | | 6.1 | | | | | 21.4 | 0.30 ± 0.03 | | | | | | | | |
| | precipitation Jerusalem | 14519 | 15.12.2014 | 709690 | 3519284 | | | 6.5 | | | | | 5.0 | 0.55 ± 0.04 | | | | | | | | |
| | precipitation Jerusalem | 14520 | 13.12.2014 | 709690 | 3519284 | | | 6.3 | | | | | 10.1 | 0.26 ± 0.02 | | | | | | | | |
| | precipitation Jerusalem | 14521 | 04.01.2014 | 709690 | 3519284 | | | 6.1 | | | | | 10.3 | 0.21 ± 0.02 | | | | | | | | |
| | precipitation Jerusalem | 14522 | 08.01.2014 | 709690 | 3519284 | | | 6.3 | | | | | 29.5 | 0.18 ± 0.02 | | | | | | | | |

The PEM is related to the exponential model (EM). Based on homogenous infiltration into a homogeneous aquifer as in (Vogel, 1967), the PEM describes mixing of those flow lines reaching the filter screen of a well. In the special case where the filter screen extends over the whole thickness of the aquifer the PEM is equivalent to the exponential model (EM), and the MRT is the only fitting parameter. The mathematical equation of DM, EM and PEM are described in Maloszewski and Zuber (1982), Małoszewski and Zuber (2002) and Jurgens et al. (2016), respectively.

### 2.3.1 Parameterization and model setup of LUMPY

To parameterize the unsaturated (vadose) zone in the recharge area, characteristic wells (Samia 2 and Herodion 1) were used (Table 2).

**Table. 2** Summary of DM, PM and PEM parameters for the wells Samia 2 and Herodion 1. Schematic graphs illustrate parameters and their individual levels in [m msl.] along both boreholes; water-filled (saturated) part of the aquifer is indicated by blue colour, screen section by checked pattern, aquitard by stripe pattern, respectively.

| | Samia 2 well | | Herodion 1 well | |
|---|---|---|---|---|
| Surface Elevation [m msl.] | 420 | | 570 | |
| Groundwater level [m msl.] | 220 | | 330 | |
| Start Filter section [m msl.] | 245 | | 316 | |
| End Filter section [m msl.] | 174 | | 237 | |
| Aquifer base [m msl.] | 150 | | 220 | |
| *LUMPY parameters* | | | | |
| Distance of screen top below water table $Z1$ [m] | 0 | | 14 | |
| Distance of screen bottom below water table $Z2$ [m] | 46 | | 93 | |
| Saturated aquifer thickness $L$ [m] | 70 | | 110 | |

*Influence of a thick unsaturated zone.* Gas tracers like CFCs and SF6 predominantly pass the unsaturated (vadose) in the gas phase, posing certain problems for their interpretation. In unsaturated zones of less than 5m thickness, the gas composition of

soil air resembles that of the atmosphere (Cook & Solomon 1995; Engesgaard et al., 2004). However, a time lag may occur

for the diffusive transport of CFCs and SF6 through thick unsaturated zones of porous aquifers (Cook & Solomon 1995). This time lag is a function of the tracer diffusion coefficients, tracer solubility in water, and moisture content (Weeks et al., 1982; Cook and Solomon, 1995). An occurring time lag always results in a gas tracer age older than the time of groundwater recharge. In fractured (or karstic) aquifers, however, the time lag may be much shorter (e.g. Darling et al., 2005) resulting in ages of CFC's or SF6 obtained from groundwater, which effectively represent residence time of groundwater since recharge

approached the groundwater table, without a time lag and as if the tracer were transported within the saturated zone only. The present study aims to estimate travel times in the thick unsaturated zone (Cook and Solomon, 1995; Plummer et al., 2006), by estimating the time lag as the relative difference in mean residence time between gas and water-bound tracers, applying both, water-bound tracers ($^3$H, $^{36}$Cl) and gas tracers (CFCs, SF6). Particularly, the wells Samia 2 and Herodion 1 of the Upper Cenomanian are selected in order to consider wells close to the recharge area with a thick unsaturated zone of 200-240 m

(Table 2). They are in the eastern part of the recharge zone (Fig. 1), where the limestone is intensely fissured vertically and partly karstified.

### 2.3.2 Well construction, aquifer data and calculation of recharge rates

The convolution integral of the partial exponential model (PEM) can be further constrained by the well construction data like (saturated) depth to the top of the screen, the screen length and saturated aquifer thickness (Jurgens et al., 2016). This data was

taken from the construction logs of the investigated production wells (Table 2).

Based on the estimated MRT of the applied lumped parameter models DM, PM and PEM, recharge rates can be estimated. However, the formulas to apply differ slightly between the different models. The following Equation (4) allows calculating the recharge rate $R$ for the PEM (Vogel, 1967; Jurgens et al., 2016):

$R = \frac{\varphi \cdot L}{MRT}$ (4)

where $\phi$ is the porosity of the aquifer, $L$ is the saturated aquifer thickness and the $MRT$ is valid for the whole aquifer (which is an output in LUMPY derived from the fitted MRT for the sample and using the well parameters Z1, Z2 and L in Table 2).

As for the DM and PM, $L$ is the distance from the groundwater surface to the depth of the centre of the screened section of the

well. Depth to centre of the water-filled screen section from groundwater level is 23 m for Samia 2 and 53.5 m for Herodion 1 (Table 2). As for the DM, not the MRT (the mean residence time of the sample) but the *peak time* is used, which is also an output of LUMPY, being calculated applying Equation (5) (Suckow, 2014b):

$Peak\ Time = \frac{MRT}{P_e}\left(\sqrt{9 + P_e^2} - 3\right)$ (5)

Tracer transport in karst areas is influenced by double porosity effects, particularly a retardation of the tracer can be assumed due to diffusive loss into the adjacent limestone of the fissures. A correction of calculated MRTs and recharge rates may be possible applying a retardation factor (Equation 6) (Maloszewski et al., 2004; Purtschert et al., 2013);

$$Retardation = \frac{\emptyset tot}{\emptyset eff} \qquad (6)$$


where $\emptyset tot$ ist the total porosity and $\emptyset eff$ is the effective porosity. An assumed porosity in the carbonate karst aquifers may vary between <2% (representing the open fissures and solution pipes) to 20% (total pore space in the aquifer rock). Groundwater recharge was also estimated using the chloride mass balance method, which was successfully applied elsewhere (Eriksson and Khunakasem, 1969; Allison and Hughes, 1978; Wood and Sanford, 1995; Purtschert et al., 2013; Crosbie et al.,

2018) and which assumes Cl-input to groundwater originates from Cl-concentration in precipitation, which becomes enriched due to evaporation only (Eq. 7):

$$R_{CMB} = \frac{P \cdot Cl_P}{Cl_{GW}} \qquad (7)$$

The formula consists of the mean annual precipitation $P$ and the chloride concentrations in precipitaion $Cl_P$ and groundwater $Cl_{GW}$ in [mg/l]. The mean annual precipitation in the recharge area is about 550 mm. The average long-term Cl-content in rain water is ca. 5 mg/l (Herut et al., 1992), while it reaches 28 mg/l and 35-38 mg/l in groundwaters of wells Herodion 1 and Samia 2, respectively.

**3 Results**

The $^{36}Cl/Cl$ ratios in precipitation, which falls in the recharge area, are assumed to be stable since the 1980s, as indicated by rainwater samples collected during winter 2014/2015, which show $^{36}Cl/Cl$ ratios of $1.3 \times 10^{-14}$ to $5.5 \times 10^{-14}$, resembling results from the early 1980s (Herut et al., 1992). Contrastingly, tritium concentration in precipitation continuously declined to about 4-6 TU today (IAEA/WMO, 2019). Well-fitting the hydrogeological trichotomy of the study area, analytical results (Table 2) resemble the regional situation and group according to the individual aquifers (Fig. 3).

*Recharge area.* In wells Samia 2 and Herodion 1, representative for the recharge area of the Upper Cenomanian aquifer, tritium concentrations of 2.1 TU and 1.2 TU as well as $^{36}Cl/Cl$ of $1.41 \times 10^{-13}$ and $1.57 \times 10^{-13}$, respectively, are observable. Further south, in well Al Reehiya, groundwater in the aquifer show lower $^{36}Cl/Cl$ and $^3H$ of $1.73 \times 10^{-14}$ and <0.5 TU, respectively. The gas tracer concentration in that part of the aquifer is low, but detectable and showed $SF_6 = 1 \pm 0.2$ fmol/l, CFC-11 = $2.3 \pm 0.3$ pmol/l, CFC-12 = $1.2 \pm 0.1$ pmol/l, and CFC-113 = $0.19 \pm 0.05$ pmol/l in Samia 2 and comparable values in Herodion 1 (Table

1). Well Jerusalem 1, which represents the Lower Cenomanian aquifer extract groundwater showing as low $^3$H concentrations (1.7 TU) as observable in the Upper aquifer, but much higher $^{36}$Cl/Cl (8.9×10$^{-13}$).

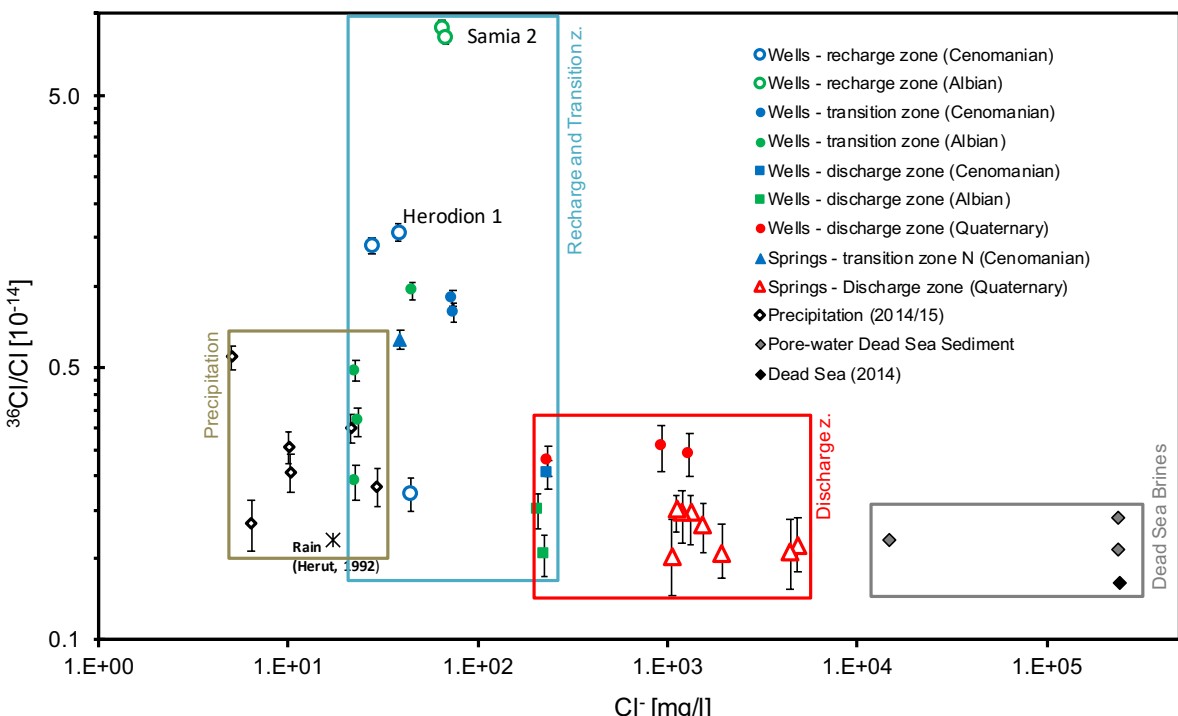

**Figure 3: Results of $^{36}$Cl/Cl versus chloride concentrations in the investigation area in log scale.**

*Transition zone.* Groundwater, emerging from the perched Turonian aquifer (Ein Auja) and from the Upper Cenomanian aquifer (Mizpe Jericho 2) show low $^{36}$Cl/Cl ratio of 6.35×10$^{-14}$ and 7.99×10$^{-14}$ and $^3$H concentrations of 2.9 TU and 1.4 TU, respectively. Well Mizpe Jericho 2 stands out due to its high CFC contents, reaching values of CFC-11 = 25 pmol/l and CFC-113 = 4 pmol/l, higher than possible in equilibrium with the atmosphere (4.5 pmol/l and 0.5 pmol/l respectively). Further south, the Arugot spring discharges in the upper Arugot Valley, about 300 m above the Dead Sea from the Upper Cenomanian aquifer, closely to the brim of the graben flank but within the transition zone. The emerging groundwater is high in SF6 (2.1 fmol/l), CFC-11 (3.8 pmol/l) and CFC-12 (1.8 pmol/l) but contains very low $^3$H (0.5 TU), suggesting a well-developed karst network which allows sufficient gas exchange of an older groundwater with recent atmosphere along its flow path.

Groundwater in the Albian aquifer (wells PWA 3, Herodion 4, Bani Naim 3) is low concentrated regarding CFCs, SF6, and $^3$H (<1 TU) and shows low $^{36}$Cl/Cl ratios of 1.9-4.9×10$^{-14}$. An exception is well Azaria 3, at the edge between recharge and transition zone, which shows higher $^{36}$Cl/Cl (9.59×10$^{-14}$) than recent precipitation, much less tritium (0.5 TU) and high concentrations of CFC-12 (5.2 pmol/l), which are again, well above values in equilibrium with the atmosphere (max 2.3 pmol/l).

*Discharge area.* Groundwater from the Upper Cenomanian aquifer (Jericho 2) show similar low $^{36}Cl/Cl$ ratios of $2.04\times10^{-14}$ and 2 TU similar to groundwater upstream in the transition zone. In groundwaters pumped in Jericho 4 and 5 from the Albian aquifer, $^{36}Cl/Cl$ and $^3H$ contents are even lower: $1.03$-$1.15\times10^{-14}$ and <0.6 TU, respectively. These low $^{36}Cl/Cl$ might result from admixing brines, which are abundant within the rift. It becomes evident in groundwater of Ein Feshkha, where interstitial brines ($^{36}Cl/Cl = 1.08\times10^{-14}$) hosted in the interstitial space of the Quaternary sediment, get leached by approaching fresh groundwaters and cause lowest observable $^{36}Cl/Cl$ of $1.01$-$1.51\times10^{-14}$.

Brackish groundwaters from the Arab wells 19-13/24A, 19-14/70 and 19-14/72, which are drilled in the Graben sediments east of Jericho, show $^{36}Cl/Cl$ of $2.29$-$2.57\times10^{-14}$ being higher than the ratios in (i) fresh groundwaters from both Cretaceous aquifers (Jericho 2, 4, 5) and (ii) in Ein Feshkha and hence, refer to a different source of salinization.

In general, the following patterns can be extracted from the hydrochemical and tracer data:

(i)      spring water of the Upper Cenomanian (Ein Auja) contains groundwater infiltrated after the era of atmospheric bomb testing, indicated by tritium and $^{36}Cl/Cl$ comparable with recent rain water;

(ii)      groundwater of the Upper Cenomanian close to the recharge area shows $^{36}Cl/Cl$ ratios higher than recent rain combined with low tritium contents, possibly referring to admixtures of water from the early fission bomb testing but before thermonuclear devices;

(iii)      along the flow path and in the lower aquifer $^{36}Cl/Cl$ is shifted to lower ratios due to admixture of saline water, the endmember being brine similar to the Dead Sea probably admixed as pore water from earlier higher sea-levels;

(iv)      the lower aquifer shows detectable admixtures of young water only in the vicinity of Jerusalem and Bethlehem, demonstrated by tritium and $^{36}Cl/Cl$;

(v)      springs in the south (Ein Feshkha and Arugot) are free of tritium and show background $^{36}Cl/Cl$ ratios indicating no recent recharge, although SF6 and CFCs are high - recent gas exchange in karst structures may have reset the SF6 and CFC "clocks".

Anthropogenic organic trace substances Simazine, NAP und ACE-K are detectable in trace concentrations in nearly all sampled wells of the Upper and Lower aquifer (Fig. 4), indicating input younger than 40 years. Combining nitrate contents with concentrations of herbicide Simazine in the sampled groundwaters allow to distinguish between the origin of $NO_3^-$, either from domestic waste water or from agriculture (Fig. 4a). All groundwaters show positive correlations between nitrate and Simazine, while there is no clear systematic trend observable, being specific for one of the aquifers nor a specific region. Studying the samples according to aquifers suggests, groundwater in the Upper Cenomanian aquifer is stronger influenced by waste water inflow than groundwaters in the Albian aquifer. Groundwater from Mizpe Jericho 2, with high $NO_3^-$ (43 mg/l) at comparable low Simazine concentrations clearly underline the contribution of waste water. Contrastingly, groundwaters from Jericho 2 and Jerusalem 1 may refer to a higher contribution from agriculture.

Looking at ACE-K and NAP as pure waste-water indicators (Fig. 4b), two distinctly opposite trends are distinguishable. Trend 1 is characterized by very low NAP concentrations and high concentrations of ACE-K as observable in wells Jericho 4 (56 ng/l) and Jericho 2 (136 ng/l). The opposite trend is found in Ein Feshkha spring D, Jerusalem 1 and Samia 2, shows high NAP concentrations (170, 168 and 137 ng/l, respectively) at low ACE-K concentrations (<20 ng/l). If one excludes Ein Feshkha, NAP concentrations decrease from recharge area (Jerusalem 1, Samia 2) to the discharge area (Jericho 2 and 4), most probably due to adsorption onto the aquifer matrix along the flow path. As for Ein Feshkha D, the NAP contamination here is comparable high as in the recharge area and much larger than the NAP contents in Ein Feshkha C and Enot Zukim. This suggests a significantly shorter residence time of the contaminant in the aquifer, which requires a source much closer to the spring, probably a leakage in the TWW-pipeline, passing the area just upstream the spring. The only groundwater without any anthropogenic contamination is Arugot spring.

## 4 Discussion

The patterns observed for the different measured substances deliver a heterogenous picture of the study area. Starting at the top, a shallow perched aquifer system with short residence times is indicated for the Ein Auja spring based on high $^{36}$Cl/Cl ratio ($6.35\times10^{-14}$), tritium content (2.9 TU) and low mineralisation like recent precipitation.

In the recharge area of the Cenomanian and Albian aquifers, clear indications for a contribution of recharge enriched with $^{36}$Cl from nuclear bomb tests is observable in freshwater wells (Cl-content <67 mg/l) Jerusalem 1, Herodion 1 and Samia 2, which show $^{36}$Cl/Cl of $1.4$-$8.9\times10^{-13}$, being much larger than in precipitation from after 1980 ($^{36}$Cl/Cl = $<5.5\times10^{-14}$). Furthermore, gas tracer results in Herodion 1 and Samia 2 imply, the aquifer contains groundwater recharged after 1960. Organic pollutants concretize it even more. High Simazine contents in Jerusalem 1 and high loads of NAP in Samia 2 and Jerusalem 1 indicate significant contamination through a young water fraction in both aquifers, the Upper Cenomanian and the Albian.

Within the transition zone, wells Al Azaria 3 (Albian) and Mizpe Jericho 2 (Upper Cenomanian) show $^{36}$Cl/Cl still larger than in precipitation from the last four decades, indicating at least similar or even older recharge periods than of groundwater in the recharge zone. However, high concentrations of CFC-11 and of ACE-K in Mizpe Jericho 2 and high concentrations of CFC-12 and NAP in Al Zaria 3 also show a significant contribution of young (waste-) water fraction to these wells. Contrastingly, anthropogenic gas tracers are close to or even below the limit of detection in groundwater samples from Albian aquifer at wells PWA 3, Herodion 4 and Bani Naim 3, indicating no fresh water input from the last decades and travel times longer than 70 years. A similar figure results from taking these three wells and forming a N-S transect through the Albian aquifer in the mountain range: their low $^{36}$Cl/Cl ratios, which are well within the range of recent precipitation, decrease from north to south while chloride remains stable. Since $^{3}$H contents are below limit of detection (<0.5 TU), the observed groundwaters are considered to be mainly pre-bomb water and hence older than 6-7 decades.

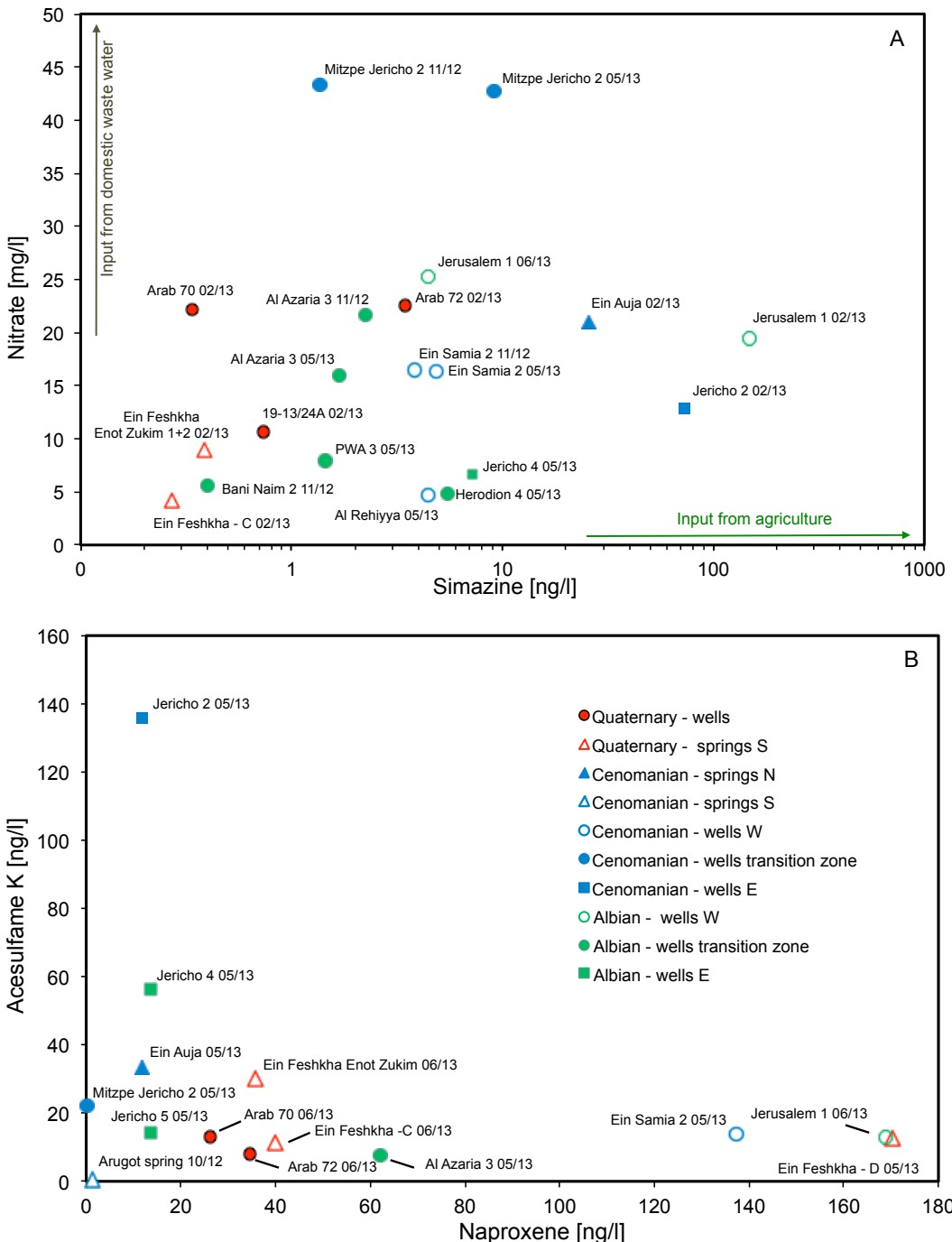

**Figure 4: showing cross plots of (A) NO₃ and Simazine in sampled waters, suggesting different sources of both pollutants; the blue and green lines bracket respectively coloured spaces and indicate the fields, in which samples from Upper Cenomanian and Albian aquifer plot; and (B) Acesulfame K and Naproxen as indicators for waste water (treated and untreated) in the sampled waters. Legend for both figures is given in (B).**

In the southern part of the study area, survey stations are even rarer. However, Well Al Rehiya shows very low $^{36}$Cl/Cl and low $^3$H, indicating either unmixed pre-bomb water or dilution of recent precipitation with much older water possibly originating from aquifer parts even further southwards. The latter is promoted by results from numerical flow modelling, which indicate an SW-NE directed upstream flow (Laronne Ben-Itzhak and Gvirtzman 2005; Gräbe et al., 2013). Negligible concentrations of NAP, ACE-K and Simazine in Arugot spring water, which emerges at the furthest end of the southern transition zone, show no anthropogenic contamination by wastewater. This leads to the assumption that open karst-conduits allow efficient exchange with the atmosphere along the subsurface flow of the spring water, resulting in CFC-11 and CFC-12 contents, in equilibrium with the atmosphere.

Groundwaters from Albian (Jericho wells 4 and 5) and from Upper Cenomanian (Jericho 2) in the discharge area are characterized by $^{36}$Cl/Cl < $2\times10^{-14}$. While the formers are free of tritium, Jericho 2 shows 2 TU, indicating at least a fraction of younger water. Since well Jericho 2 is drilled directly at the mouth of the Qilt Valley, we assume, the young water fraction may reach the well through rapid infiltration through fractures and karst conduits within the Valley. The young water fraction in Jericho 2 is also proven by high Simazine and ACE-K concentration in the groundwater of that well.

Low $^{36}$Cl/Cl ratios in the wells drilled into the Quaternary section east of Jericho (ca. $2\times10^{-14}$) and in the Ein Feshkha springs (ca. $1\times10^{-14}$) refer to enrichment of fresh groundwater with different sources of salinity, showing low $^{36}$Cl/Cl ratios. The brackish wells East of Jericho are known to draw groundwater from the Upper Cenomanian aquifer (Khayat et al., 2006a), which passes the graben fault and enters into the sediment body, where it may leach abundant evaporitic minerals. It also contains water from agricultural irrigation and waste water as confirmed by high $NO_3$ concentrations, the presence of Simazine and occasionally remarkable ACE-K contents.

Contrastingly, Dead Sea brines, which get released from the sedimentary body, mix in Ein Feshkha with approaching fresh groundwaters from the Graben shoulder. The low $^3$H (<1 TU) in Ein Feshkha indicate the age of the admixing Dead Sea brine to be before 1960. During that time, lake level was at -390 m msl. and higher (Hassan and Klein, 2002). At the time, Dead Sea brine infiltrated into the shallow lake bed, where nowadays the observed springs emerge, at an elevation of -392 to -395 m msl. However, Ein Feshkha springs C, D and Enot Zukim receive young water fractions, since NAP and ACE-K contents in the former are as high as in groundwater in the recharge zone (Jerusalem 1) and in the latter at least enhanced.

The next section will exemplify a more detailed modelling of mean residence times with transient tracers where the data allows this approach. This was possible only on two wells. Wells of the Albian aquifer in the transition zone are not interpreted using LPM due to contaminations with CFCs (e.g. Al Azaria 3, Table 1) or very low concentrations of the anthropogenic gases (e.g. Bani Naim 3). If the measured concentrations of the anthropogenic trace gases CFCs and SF6 are much higher than expected from solubility equilibrium with the atmosphere this is regarded as contamination (e.g. Mizpe Jericho 2). The tritium concentrations of 1.5 TU and 2 TU in Mizpe Jericho 2 and Jericho 2, respectively, indicate a reasonable fraction of recent rain.

Due to the continuous outflow of the springs along the coastline and chemical mixing patterns of the spring water, it is assumed that there is also a connection to the water resources of the Lower Cretaceous aquifer, which could provide water with longer residence times. All springs in the discharge area (e.g. Ein Feshkha) have less than 0.5 TU and $^{36}$Cl/Cl <$1.5\cdot10^{-14}$. However,

the organic tracers ACE-K, Simazine and NAP are detectable also in these springs. Their values in Ein Feshkha are between 5 to 30 times above the detection limit (Table 1 and section 2.2). Since the input function of the organic tracers cannot be

quantified, the fraction of young water, representing that concentration cannot be calculated. This discrepancy (no tritium, no anthropogenic $^{36}$Cl, but organic tracers) can be understood by quantifying the detection limit of "young water" that tritium allows: recent rain in the area has 4 TU, so a value of <0.5 TU is equivalent to less than 12% recent rain. This implies the young end member in the mixture must have a concentration of ACE-K, Simazine® and NAP >10 times higher than measured in the groundwater samples. A more precise quantification of the young water fraction in these springs is only possible with

either a lower detection limit for tritium (e.g. determination of $^3$H via ingrowth with detection limit 0.005 TU (Bayer et al., 1989; Beyerle et al., 2000; Sültenfuss et al., 2009) or a precise quantification of the input of ACE-K, Simazine® and NAP. Both is beyond the scope of the present study.

## 4.1 Lumped parameter modelling: delay of gas tracer in the unsaturated zone

Detailed lumped parameter modelling of tritium, $^{36}$Cl/Cl and the gas tracers was performed only for the wells Samia 2 and

Herodion 1 since only for these there is a consistent data set for all tracers (Table 1). Earlier interpretation of groundwater hydraulics based on groundwater level measurements determined a very fast transfer velocity of the water phase through karst holes to the groundwater table (Jabreen et al., 2018). LPM was therefore done in several steps: first a mean residence time in agreement with measured values for tritium and the bomb-spike of $^{36}$Cl/Cl was derived. Then a delay for the gas tracer CFCs and SF6 was derived assuming a simple piston flow transport to describe the residence time of these tracers in the unsaturated

zone. Once the delay created an agreement between the water – bound tracer tritium and $^{36}$Cl/Cl with the gas tracer CFC-11, CFC-12, CFC-113 and SF6, the mean residence time in the saturated zone was investigated using the PM, DM and PEM to describe flow in the saturated zone. From the MRT in the saturated zone, groundwater recharge was estimated in a final step and compared with results from the CMB method and earlier numerical groundwater models.

Results for tritium and $^{36}$Cl/Cl fit to MRT of about 10a in the saturated zone using the PM, the dispersion model results in 20a

MRT and for the partial exponential model 16 and 20 years MRT are estimated (Table 4). These estimated MRT of the water-bound tracers then allowed for determining the gas delay in the unsaturated zone. For every applied model (PM, PEM and DM) the gas delay is estimated using the calculated model curve of the water-bound tracers and using the delay as parameter to fit the gas tracer measurements of Samia 2 and Herodion 1. This resulted in specific gas delays for every concentration of the CFC's and SF6, exemplified for CFC-12 in Figure 5 and provided for all gas tracers in Table 3.

Estimates for every model result in different gas delays related to the MRT of water-bound tracers in the wells Samia 2 and Herodion 1 (Table 3). The delays of the CFC's are comparable to each other and differ from 14 to 24 years for Samia 2 (Fig. 5) and from 14 to 30 years for Herodion 1 over all calculated models. The gas delay in Samia 2 is lower than in Herodion 1, which is reasonable regarding the thicker unsaturated zone in Herodion 1 (Table 2). Ideally the transport of gas tracers in the subsurface would result in similar delays for all gas tracers (Cook and Solomon, 1995). The delay determined for SF6 was

significantly lower than for the CFC's (Table 3) indicating higher concentrations of SF6 as compared to the expectation from

the CFC model results. This could be a result of biological degradation of the CFC's in the upper part of the unsaturated karst zone, or from considerable excess air, which influences SF6 much more than the CFCs. A decision between these two processes is possible determining excess air independently measuring the concentrations of all noble gases, which were not available in the present study.

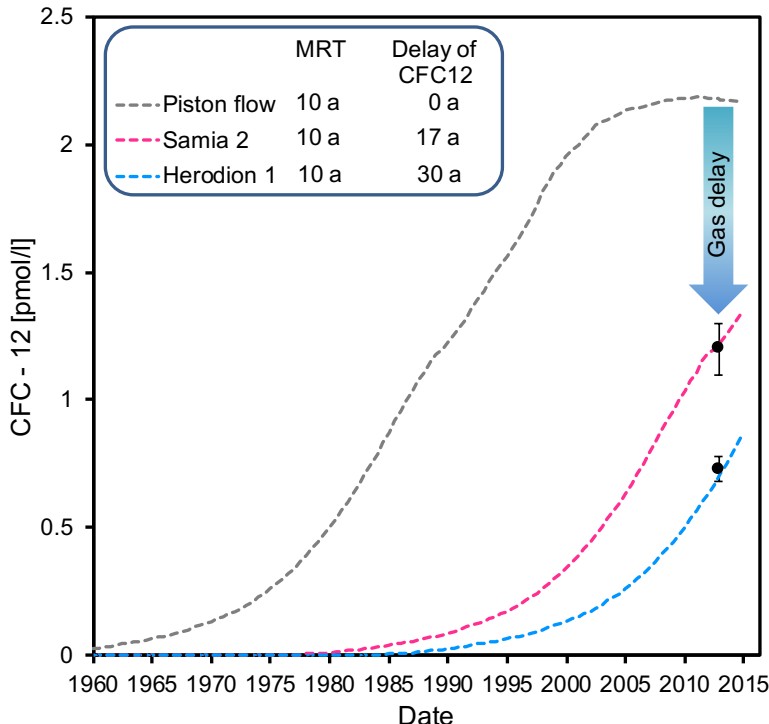

**Figure 5: In a CFC-12 vs. time plot, a piston flow model with MRT of 10 years obtained from the water-bound tracer tritium and [36]Cl/Cl fits the CFC-12 measurements with a gas delay of 17 years for Samia 2 and 30 years for Herodion 1.**


**Table 3.** Summary of gas delay in the unsaturated zone for each gas tracer and the different models. PM = piston flow model, DM = dispersion model, PEM = partial exponential model. For the parameters of the PEM see Table 2.

| Model | PM | | DM | | PEM | |
|---|---|---|---|---|---|---|
| Parameters | Samia 2 | Herodion 1 | Samia 2 | Herodion 1 | Samia 2 | Herodion 1 |
| CFC-11 | 24 | 28 | 14 | 18 | 20 | 25 |
| CFC-12 | 17 | 30 | 14 | 20 | 20 | 28 |
| CFC-113 | 23 | 22 | 9 | 14 | 17 | 22 |
| SF6 | 8 | 8 | 0 | 0 | 7 | 6 |

## 4.2 Lumped parameter modelling: mean residence times in the saturated zone

In the next step, the determined gas delays of the gas tracers (Table 3) in the unsaturated zone were used to estimate the MRT of the water-bound tracers and the gas tracers in the saturated zone. Only the gas delay allowed a fit to the measurement combination of the gas tracer concentrations and the water tracer results of Samia 2 in the time range of 10 to 15 years MRT for the piston flow model, 17 to 28 years MRT for the dispersion model and 26 to 46 years for the partial exponential model (Table 4). Comparing the different model approaches, estimates of the saturated MRT systematically increase in the sequence

PM-DM-PEM in both wells, which is a result of the increasing amount of dispersion and mixing in these models.

In contrast to Samia 2, it was not possible to fit the results of Herodion 1 without an admixture of tracer-free old water. With and without delay the lumped parameter curves do not fit the measurements of the lower tritium concentration in Herodion 1 (1.2 TU) whereas they do for Samia 2. The well Herodion 1 is situated in the recharge zone, but also in the lower part of the Upper Cenomanian aquifer. The model results therefore indicate an admixture of at least one older water component ascending

from the Albian aquifer and diluting the younger tritium concentrations. This can be described with a two-endmember mixing line, which fits the tritium measurement of Herodion 1 (Fig. 6).

**Table 4** Summary of modelled MRTs of Samia 2 and Herodion 1 extracted from different tracer combinations. PM = piston flow model, DM = dispersion model, PEM = partial exponential model.

*estimated fit.

| Sampled well | Samia 2 | | | Herodion 1 | | |
|---|---|---|---|---|---|---|
| Model | PEM | DM | PM | PEM | DM | PM |
| CFC-11 vs CFC-113 | 31 – 44 | 20 - 23 | 10 - 14 | 17 - 22 | no fit | 11 - 13 |
| CFC-11 vs. SF6 | 31 – 44 | 21 - 25 | 9 - 13 | 21 - 23 | 17 - 22 | 10 - 12 |
| CFC-12 vs. SF6 | 38 – 46 | 21 - 23 | 10 - 13 | 18 - 20 | 22 - 24 | 10 - 11 |
| $^{36}$Cl/Cl vs. SF6 | 36 – 37 | 20 - 22 | 9 – 15* | 17 - 18 | 19 - 20 | 9 - 15 |
| $^{36}$Cl/Cl vs. CFC-113 | 36 – 38 | 21 - 22 | 11 – 15* | 16 - 28 | no fit | 9 - 18 |
| $^{36}$Cl/Cl vs. CFC-12 | 36 – 38 | 21 - 23 | 11 – 14* | 17 - 18 | 20 - 21 | 11 - 12 |
| $^{36}$Cl/Cl vs. Tritium | 36 – 38* | 20 - 22 | 9 - 29 | | Mixing ratios of 3:7 to 4:6 (1 a MRT and 170 a MRT) | |
| Tritium vs. SF6 | 26 – 44 | 18 - 23 | 10 - 15 | | Mixing ratios of 3:7 to 4:6 (1 a MRT and 170 a MRT) | |
| Tritium vs. CFC-12 | 38 – 45* | 21 - 23 | 10 - 14 | | Mixing ratios of 3:7 to 4:6 (1 a MRT and 170 a MRT) | |


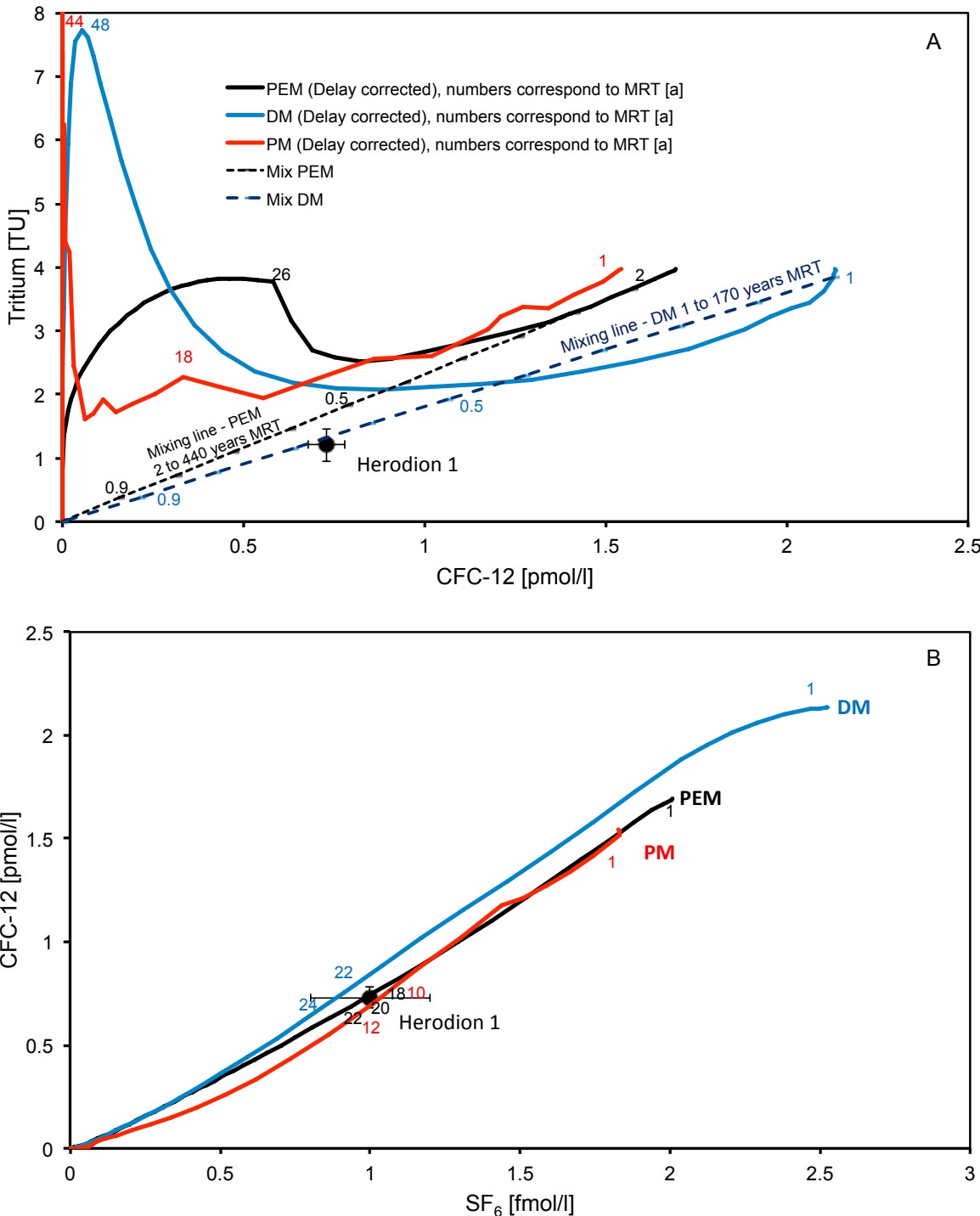

**Figure 6: Modelling results for groundwater from Herodion 1 well, applying PM (red), PEM (black) and DM (blue) and (A) tritium vs. CFC-12 and (B) CFC-12 vs. SF6. The calculated delays of CFC-12 are according 30 yrs., 20 yrs., and 28 yrs., to the models PM, DM and PEM, respectively (ref. Table 3).**

## 4.3 Recharge rates and dual porosity

The mean residence times resulting from lumped parameter models PM, DM and PEM in Table 4 were used to calculate recharge rates according to Equations (4) and (5). Within a single LPM the MRT have uncertainties in the range of 50% and this uncertainty increases to approximately a factor of 2 if all models are regarded as equally probable (Table 4). Therefore, a major uncertainty for the calculation of recharge results from the unknown porosity of the karst aquifer, which is estimated to be somewhere between 2% and 20%, creating an additional uncertainty of a factor of 10. It is therefore useful to consider whether this uncertainty can be further restricted.

Recharges rates from chloride mass balance (CMB) highly depend on the value used for chloride in precipitation since the chloride concentration in the Cenomanian aquifer for youngest groundwater (tritium >1.5TU) varies only by 22% between 28 mg/l and 44 mg/l (Table 1). The Cl$^-$ values measured in precipitation during the present study vary between 5 and 30 mg/l resulting in a spread between 62 mm/a and 375 mm/a for the recharge derived from CMB according to Equation 7, assuming an average precipitation amount of 550 mm/a. This is comparable to recharge rates used in numerical flow modelling studies of the eastern groundwater catchment at the Dead Sea, who estimated 124-292 mm/a (Guttman, 2000) and 100-300 mm/a (Gräbe et al., 2013) in the mountainous recharge area.

Tracer-derived recharge values compare best with the recharge estimates from CMB and numerical modelling if a porosity of 10% is assumed. The tracer-derived values for 20% porosity are in the range of 40% to 100% of the mean annual precipitation of Jerusalem and especially the latter value seems unrealistically high. The highest values originate from the PM and the PEM. The PM neglects mixing and retardation along the flow path completely, which of course is the least probable assumption. The PEM, however, considers mixing in the well, but it relies on the flow system being extremely homogeneous, as in the "Vogel" aquifer (Vogel, 1967; Jurgens et al., 2016). A few preferential flow paths can easily disturb this picture towards smaller MRT and thus larger recharge values. An indication why the tracer-derived recharge values using 2% porosity are unrealistically low comes from the hydrogeology of the karst aquifer itself. The 2% porosity can be attributed to structural fractures and macropores in the limestone, which account for rapid reaction of the groundwater surface to precipitation events (Jabreen et al., 2018). However, according to Equation (5) the tracers are not expected to indicate this flow but they would be retarded by diffusion of tracer into and out of the rock matrix (Maloszewski et al., 2004; Purtschert et al., 2013; Suckow et al., 2019). This means the tracers never "see" only the 2% fracture and macropore space of the karst but also at least a part of the total rock matrix with its total porosity of rather 20%. If the effective porosity in the karst system is 2% and the total porosity is 20%, the retardation factor according to equation 5 would be 10. Applied to the lumped parameter models, this retardation means that the calculated MRT are too old and the calculated recharge rates with 2% porosity are too low.

**Table 5.** Results of recharge rates [mm/a] based on maximum and minimum of the estimated MRT [a] obtained from PM, DM and PEM for Samia 2.

| Model | MRT | 2% porosity Recharge [mm/a] | 10% porosity Recharge [mm/a] | 20% porosity Recharge [mm/a] |
|---|---|---|---|---|
| **PM** | MRT 9 a | 51 | 256 | 511 |
| | MRT 29 a | 16 | 157 | 315 |
| **DM** | MRT 16.3 a | 28 | 141 | 282 |
| | MRT 22.6 a | 20 | 120 | 203 |
| **PEM** | MRT 31 a | 45 | 226 | 452 |
| | MRT 45 a | 31 | 156 | 311 |

## 5 Conclusions

The present study derived groundwater flow patterns, mixing end-members, transport times and recharge estimates in the upper and lower Cretaceous aquifers of the EAB between the recharge area around the central mountain ridge and the discharge zones close to the lower Jordan Valley and the Dead Sea, shared by Israel and Palestine. This was possible, despite a very low number of measurements and a complicated karst hydrogeological setting, using a powerful combination of multiple lines of evidence from hydrogeology, hydrochemistry, anthropogenic organic trace substances and classical environmental age-dating tracers like tritium, CFCs, SF6 and $^{36}$Cl/Cl. The present study is one of presently only a handful demonstrating the useful application of atmospheric bomb-test derived $^{36}$Cl to study groundwater movement. The lower Cretaceous aquifer was found to be basically free of tritium and other anthropogenic environmental tracers like CFCs, SF6 and bomb-derived $^{36}$Cl and, therefore, has groundwater transport times larger than 50 years. Groundwater in the upper Cretaceous aquifer contains anthropogenic trace substances, indicating time scales of groundwater flow of a few decades and shows clear indications of preferential flow paths as expected for karstified groundwater systems. In two springs the combination of several environmental tracers ($^3$H, bomb-derived $^{36}$Cl, CFC-11, CFC-12, CFC-113, SF6) allowed estimates of the mean residence time in the saturated zone, gas transfer delays in the unsaturated zone and mixing ratios with older groundwater. These estimates were confirmed by anthropogenic substances from agriculture (nitrate and pesticides like Simazine®), pain killers (Naproxen®) and sweeteners (Acesulfame K®). Calculated recharge rates based on MRT estimates compare well with chloride mass balance and numerical flow modelling. The multi-tracer methodology presented here is applicable in other data sparse areas with complex hydrogeology (karst or fractured) with or without anthropogenic influence.

## Acknowledgments

The authors are grateful for the support of the project, which was partly covered by the SUMAR project, funded by the German Ministry of Education and Research (grant code: 02WM0848) and by the DESERVE Virtual Institute (grant code VH-VI527) funded by the Helmholtz Association of German Research Centres. We wish to thank Y. Yechieli (GSI) for hosting and fruitful discussions, A. Avalon (GSI) for sampling recent precipitation, J. Guttman (Mekorot) for fruitful discussions and access to Mekorot wells, M. Hadidoun (PWA) for enabling sampling at PWA wells. We also thank the Israel Hydrological Survey for providing support and well data. Parts of this research was carried out at the Ion Beam Centre at the Helmholtz-Zentrum Dresden-Rossendorf e.V., a member of the Helmholtz Association, . We hence thank S.M.E. Baez, R. Ziegenrücker and the DREAMS operator team for supporting the AMS-measurements. The authors thank J. Guttman and N.A. Sheffer for their constructive reviews and E. Morin for efficiently handling the manuscript.

*Data availability.* The data on which the study bases are given in the manuscript (Table 1).

*Author contributions.* CW, TR, BM and CS outlined and conducted field work. SM, SP, GR conducted AMS-measurements. CW, ASu, CS analyzed the data. BM, TR, UM, CM, ASa and SMW were involved in discussion process. CW, ASu and CS wrote the paper. All authors revised the paper and approved the final version.

*Competing interests.* The authors declare that they have no conflict of interest.

*Special issue statement.* This article is part of the special issue "Environmental changes and hazards in the Dead Sea region (NHESS/ACP/HESS/SE inter-journal SI)". It is not associated with a conference.

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
