# Peer review of "A multi-environmental tracer study to determine groundwater residence times and recharge in a structurally complex multi-aquifer system"

_Hydrology and Earth System Sciences, 2019_

## Referee Comment (RC1) · Joseph Guttman (Referee) · 8 Oct 2019

The paper is very interesting because it is the first time that a research in this region is using these tracers as an additional tool that improve the hydrological knowledge of this complicated aquifer. The results are compatible to the well operating data. For example, the results in Mizpe Jericho 2 that show contamination of sewage that we know it is coming from leakage from the sewage pipe passing near the well. I warmly recommend publishing the paper as is.

---

## Referee Comment (RC2) · Nathan A. Sheffer (Referee) · 5 Nov 2019

The authors demonstrate how groundwater can be dated using natural and anthropogenic tracers. I find the paper very interesting and with high academic value. The authors fail to site important work dealing with this aquifer, esp. groundwater flow rates and direction (LL Ben-Itzhak, 2005 - Groundwater flow along and across structural folding: an example from the Judean Desert, Israel). According to this work, groundwater flow is not W-E but rather SW-NE. Therefore, the springs mentioned in this work do not represent water infiltrating west of them, but rather SW of them. Another important issue is local contamination (as mentioned by Joseph Guttman in his comment to

this paper). How do the authors differentiate between local contamination by a burst sewage pipe adjacent to a well or spring, from contamination infiltrating in the recharge area? Other comments: line 14 and 63: I think that 5 million people is an over estimation for the population using this aquifer. line 66: Two references missing from list. line 68-69: Two references missing from list. line 85: Dead Sea Rift, not Jordan... line 106: groundwater does not easily dissolve anhydrite and aragonite. Figure 1: Site 23 on map should be 24.

Chlorine source according to this paper is rainwater. what about Cl from soil? esp. fertilizers such as KCl? line 334 and 338: groundwater is ploral, no need for "s". line 336: Embracing - this is the wrong word for this.

The paper need much more proofing. Ex.: in some places Table and in others Tab.; in some places Figure and in others Fig.; SF6 -some places the "6" is in subscript, and some not; 36CL -some places the "36" is in superscript, and some not;

---

## Author Comment (AC1) · 7 Nov 2019

Dear Dr. Guttman, we are pleased to receive your very positive comment and the recommendation to publish our paper in HESS. With kind regards and on behalf of all authors, C. Siebert

---

## Author Comment (AC2) · 7 Nov 2019

Dear Mr. Sheffer, we very much appreciate your interactive comment on our manuscript and would like to reply to your comments.

We thank you for your indication of missing important citations such as Laronne Ben-Itzhak and Gvirtzman 2005 and included it into the manuscript. We also carefully reviewed the manuscript for given statements as for groundwater flow directions. We absolutely agree with Laronne Ben-Itzhak and Gvirtzman and also with the results of (Gräbe et al., 2017; Hydrological Processes), who postulate the groundwater flow direction to be NE-oriented. We do not doubt about and explicitly added a respective

formulation. When describing flow within the EAB, it is generalized following gravity and the overall W-E direction, given by a N-S oriented mountain range, where the recharge occurs and a parallel oriented discharge area: the N-S trending Jordan-Dead Sea Valley. However, the presented study is not intended to give indications for flow-directions, which would be beyond the possibilities of the applied tracers. The geographic location of recharge areas of particular wells and springs is only (if at all) detectably applying numerical flow models, which are well calibrated using water table information and geochemical and isotopic trace information, such as rare earth pattern. Contamination of groundwater has been detected by organic tracers, which give indications for sources of pollutants. To differentiate between local and remote pollution sources is a difficult issue. However, indications may be carved out by the different behaviors of organic substances (affinity to adsorb, metabolism by microbes, inert traveling) in combination with water-bound and gas-bound tracer ages. for example, since NAP has the tendency to get immobilized during transfer through the aquifer by adsorption, sampling locations such as Ein Feshkha must have very low concentrations, if one expects the input in the remote recharge area. This is true for all springs in Enot Zukim, except Ein Feshkha D, which indicates a local contamination because: NAP is remarkably high and contamination is focussed to that spring. As for your comment to the number of inhabitants, who is reliant to the EAB you are most probably right and we defused the expression. Thank you for pointing to missing references, which we included in the list.

In geo-scientific literature, the term Jordan Dead Sea Rift is common, although a huge variety of terms exist to describe the active boundary between the two plates. However, we changed the term and deleted "Jordan".

The dissolution of evaporite minerals is doubtless different starting with easily soluble halite to less easy soluble aragonite. However, the "intense dissolution" of all the evaporite minerals in the Dead Sea Sediments is existing, since microbial produced H2SO4 intensifies the ability of pure water to dissolve these minerals (see Ionescu et al. 2012; PLoS One).

As for chlorine as input source, we assume on the long run, the major source of Cl in recharge is rainwater, which comes loaded with salinity from the Mediterranean Sea. Even if salt is precipitated at the soil due to evaporation, it is either directly precipitated from infiltrated precipitation or from irrigation water, which is groundwater and hence owns similar isotope signatures. Fertilizers containing potasium do host it either as oxide (K2O) or nitrate (KNO3), rather than as cloride, since its application would increase the speed of soil degradation. The large brands in Israel (ICL and Haifa Group) do offer K-hosting fertilizers in the given form and also fertilizers used in the Westbank are composed in that way (e.g. UNCTAD report 2015). We hence excluded potash from our considerations.

As for the typos, we thank you very much and you are right about the heterogeneity of notations of SF6, 36Cl, Table and Figure. We corrected it.

On behalf of all authors, I thank you again for reviewing the manuscript, C. Siebert

---

## Author Response (AR1)

Nathan A. Sheffer (Referee)
nathan.sheffer@icl-group.com

Dear Mr. Sheffer, we very much appreciate your interactive comment on our manuscript and would like to reply to your comments.

**The authors demonstrate how groundwater can be dated using natural and anthropogenic tracers. I find the paper very interesting and with high academic value.**

**The authors fail to site important work dealing with this aquifer, esp. groundwater flow rates and direction (LL Ben-Itzhak, 2005 - Groundwater flow along and across structural folding: an example from the Judean Desert, Israel). According to this work, groundwater flow is not W-E but rather SW-NE. Therefore, the springs mentioned in this work do not represent water infiltrating west of them, but rather SW of them.**

We thank you for your indication of missing important citations such as Laronne Ben-Itzhak and Gvirtzman 2005). We included it into the manuscript.
We also carefully reviewed the manuscript for given statements concerning groundwater flow directions. We do not criticize that study it is not intended to discuss it within the manuscript, however more recent research, such as Sachse et al. (2017) in HP, show NE orientation is valid for the southern portion of the drainage area, while the northern drainage area forms E- and ESE-oriented flow.
In the presented study, flow within the EAB is generalized and given by a N-S oriented mountain range, where the recharge occurs and a parallel oriented discharge area: the N-S trending Jordan-Dead Sea Valley. However, the presented study is not intended to give indications for flow-directions, which would be beyond the possibilities of the applied tracers. The geographic location of recharge areas of particular wells and springs is only (if at all) detectably applying numerical flow models, which are well calibrated using water table information and geochemical and isotopic trace information, such as rare earth pattern.

**Another important issue is local contamination (as mentioned by Joseph Guttman in his comment to this paper). How do the authors differentiate between local contamination by a burst sewage pipe adjacent to a well or spring, from contamination infiltrating in the recharge area?**

Contamination of groundwater has been detected by organic tracers, which give indications for sources of pollutants. To differentiate between local and remote pollution sources is a difficult issue. However, indications may be carved out by the different behaviors of organic substances (affinity to adsorb, metabolism by microbes, inert traveling) in combination with water-bound and gas-bound tracer ages. for example, since NAP has the tendency to get immobilized during transfer through the aquifer by adsorption, sampling locations such as Ein Feshkha must have very low concentrations, if one expects the input in the remote recharge area. This is true for all springs in Enot Zukim, except Ein Feshkha D, which indicates a local contamination because: NAP is remarkably high and contamination is focused to that spring.

**Other comments:**

**line 14 and 63: I think that 5 million people is an over estimation for the population using this aquifer.**

As for your comment to the number of inhabitants, who is reliant to the EAB you are most probably right and we defused the expression.

**line 66: Two references missing from list.**
Thank you for pointing to missing references, which we included in the list.

**line 68-69: Two references missing from list.**
Thank you for pointing to missing references, which we included in the list.

**line 85: Dead Sea Rift, not Jordan...**
In geo-scientific literature, the term Jordan Dead Sea Rift or Transform is common, although a huge variety of terms exist to describe the active boundary between the two plates. We changed the term and deleted "Jordan".

**line 106: groundwater does not easily dissolve anhydrite and aragonite.**
The dissolution of evaporite minerals is doubtless different starting with easily soluble halite to less easy soluble aragonite. However, the "intense dissolution" of all the evaporite minerals in the Dead Sea Sediments is existing, since microbial produced $H_2SO_4$ intensifies the ability of pure water to dissolve these minerals (see Ionescu et al. 2012; PLoS One).

**Figure 1: Site 23 on map should be 24.**
It is corrected.

**Chlorine source according to this paper is rainwater. what about Cl from soil? esp. fertilizers such as KCl?**
As for chlorine as input source, we assume on the long run, the major source of Cl in recharge is rainwater, which comes loaded with salinity from the Mediterranean Sea. Even if salt is precipitated at the soil due to evaporation, it is either directly precipitated from infiltrated precipitation or from irrigation water, which is groundwater and hence owns similar isotope signatures. Fertilizers containing potassium do host it either as oxide ($K_2O$) or nitrate ($KNO_3$), rather than as chloride, since its application would increase the speed of soil degradation. The large brands in Israel (ICL and Haifa Group) do offer K-hosting fertilizers in the given form and also fertilizers used in the Westbank are composed in that way (e.g. UNCTAD report 2015). We hence excluded potash from our considerations.

**line 334 and 338: groundwater is ploral, no need for "s".**
corrected

**line 336: Embracing - this is the wrong word for this.**
This is correct, we exchanged it.

**The paper need much more proofing: Ex.: in some places Table and in others Tab.; in some places Figure and in others Fig.; SF6 -some places the "6" is in subscript, and some not; 36CL -some places the "36" is in superscript, and some not;**
We thank you for indicating the heterogeneity of notations of $SF_6$, $^{36}Cl$, Table and Figure. We corrected it.

[revised manuscript text omitted]